# Host Recognition and Specific Infection of *Endomelanconiopsis endophytica* during Early Infection

**DOI:** 10.3390/jof9101040

**Published:** 2023-10-23

**Authors:** Yan Xie, Liuqing Shi, Keke Cheng, Yang Li, Shixiao Yu

**Affiliations:** Department of Ecology, School of Life Sciences/State Key Laboratory of Biocontrol, Sun Yat-sen University, Guangzhou 510275, China; xiey45@mail2.sysu.edu.cn (Y.X.); shilq3@mail2.sysu.edu.cn (L.S.); chengkk@mail2.sysu.edu.cn (K.C.); liyang233@mail2.sysu.edu.cn (Y.L.)

**Keywords:** comparative transcriptomics, effector, genomics, specific infection, whole-genome sequence

## Abstract

Coevolution between the pathogen and host plant drives pathogenic effector diversity. However, the molecular mechanism behind host-specific pathogenesis remains to be explored. Here, we present a 43 Mb whole-genome sequence of *Endomelanconiopsis endophytica* strain LS29, a host-specific pathogen of the common subtropical tree *Castanopsis fissa*. We described its genome annotations and identified its effector candidates. By performing temporal transcriptome sequencing of *E. endophytica* on *C. fissa* during early infection, we found that *E. endophytica* repressed other microbes in order to attack the tissue of the host by producing antibiotics earlier than 24 h post-inoculation (hpi). Simultaneously, a variety of effectors were secreted to recognize the host plant, but most of them showed a significantly opposing expression regulation trend after 24 hpi, indicating that 24 hpi represents a key time point between host recognition and specific infection. Furthermore, a comparison of isoenzymes showed that only a few effectors were identified as specific effectors, which were involved in hydrolyzing the compounds of the plant cell wall and releasing fatty acids during the early infection of *C. fissa*. Our results determined host recognition timing and identified a specific catalog of effectors, which are crucial for revealing the molecular mechanism of host-specific pathogenesis.

## 1. Introduction

The Janzen-Connell hypothesis suggests that the maintenance of plant diversity is driven by density- and distance-dependent in tropical forests, and host specialization is a crucial assumption of this hypothesis [1,2]. It has demonstrated that host-specific pathogens have a stronger effect on promoting plant diversity compared to general pathogens [3]. Previous studies on the pathogenic mechanism of host-specific pathogens mainly focused on monoculture crops [4], and how host-specific pathogens infect host trees in forest will contribute to understanding the maintenance of forest diversity. In 2020, we isolated *Endomelanconiopsis endophytica* strain LS29 from the common subtropical tree *Castanopsis fissa*, and its host-specific pathogenicity was determined [5], but the pathogenesis of host-specific pathogen *E. endophytica* on *C. fissa* remains unclear.

Botryosphaeriaceae fungi are widely distributed endophytes across the world, but most of them represent potential pathogen threats to woody plants [6]. The life cycle of Botryosphaeriaceae can be divided into an endophytic phase and a pathogenic phase. The endophytic phase may be an incubation period where they may survive for many years as endophytes and then switch into a pathogenic lifestyle, causing diseases such as leaf spots, fruit and root rot, cankers, and dieback [7]. As a member of this family, *E. endophytica* produces a dark green to nearly black colony on potato-dextrose agar [8]. 

Many phytopathogens are generally recognized, and defended against, by plants and are only able to infect certain plant species, which determines their host ranges [9]. The pathogenic mechanisms of fungi are complex and various. By contrast, plants generally have two defense systems to detect pathogens: One is to recognize conserved pathogen-associated molecular patterns (PAMPs) by pattern recognition receptors (PRRs) activating the defense system, referred to as PAMP-triggered immunity (PTI) [10]; and the other is effector-triggered immunity (ETI), in which fungal specific avirulence proteins are recognized by plant-specific resistance proteins, resulting in disease resistance [11]. However, fungal virulence proteins successfully escape recognition by the plant, eventually leading to the development of disease. The “Zigzag model” has been proposed to explain the interactions between plants and pathogens: to overcome PTI, pathogens secrete effectors, and to suppress ETI, pathogens diversify the recognized effectors [12,13].

Effectors are cytotoxic or otherwise damaging plant cells in the apoplast or cytoplasm [14]. The known effector secretion systems in bacteria contain six secretion systems (types I–VI) of Gram-negative bacteria and the unique mycobacterial type VII secretion system [15]. Unlike bacteria, the effectors of fungi are various and diverse, and there is no universally approved classification for the secretion system as of yet. Effector genes have evolved multiple ways to evade host recognition in a spatial and temporal manner depending on the stage of infection, such as through disrupting physical barriers, creating conditions conducive to invasion, masking microbial features, inhibiting the activity of PRR complexes, interfering with PRR translation, and inhibiting the activity of plant proteolysis [16,17]. For example, *Sclerotinia sclerotiorum* breaks down a wide range of complex plant polysaccharides by cell wall degrading enzymes (CWDEs) [18]. lysin motif (LysM) proteins are the most extensive class of apoplastic effector proteins, which sequester chitin fragments released to evade the perception of chitin fragments as PAMPs by plant PRRs during infection [19]. A microRNA-like RNA (bba-milR1) is exported to hijack the host RNA-interference machinery by *Beauveria bassiana* [20]. Protein essential during penetration 1 (Pep1) acts as an inhibitor of host peroxidases, suppressing the apoplastic oxidative burst during the infection of maize by *Ustilago maydis* [21]. After hundreds of years, the coevolution between plants and pathogens has resulted in specific effectors to circumvent host immunity, thereby forming specific pathogeneses. *Alternaria alternata* strains produce a variety of host-selective toxins (HSTs) that cause disease in specific hosts [22].

At present, more than 1500 fungal genomes have been collected in the Ensembl Fungi Release 56 database (https://fungi.ensembl.org/index.html, accessed on 13 June 2023). Based on the fungal genome, a variety of bioinformatics methods are used to predict effectors, which are important virulence factors of pathogenic fungi [14]. In general, the criteria of candidate secretory effector proteins include a signal peptide, no trans-membrane domains, no glycosylphosphatidylinositol (GPI) lipid anchoring site, extracellular localization, and, often, species specificity [23,24]. More than 100 tools including pipelines and online servers are available to identify the conserved motifs, characteristic sequences, and structural features of proteins, and multiple tools can be combined for the genome-wide identification of effector proteins [24]. The ability of Botryosphaeriaceae species to cause disease may rely on the production of effectors, such as cell-wall-degrading enzymes and degrading wood parietal compounds [25].

The transcriptome of fungal genes in temporally infected hosts reveals further details about their pathogenic processes and mechanisms. The genes encoding glycosyl hydrolases, cutinases, and LysM domain-containing proteins have been shown to be upregulated in *Magnaporthe oryzae* after inoculation on rice [26]. A total of 307 effector candidates of *Leptosphaeria maculans* were identified and over-expressed in the infected stems of *Brassica napus*, suggesting that they are involved in systemic colonization [27]. In addition, differences in gene expression may reveal the determinants of virulence variation among *Zymoseptoria tritici* strains during the infection of wheat [28].

Currently, few previous data on *E. endophytica* are available. Therefore, our knowledge of specific effectors of *E. endophytica* is of vital importance to understanding the molecular mechanisms underlying how *E. endophytica* invade host plants. In this study, the first whole-genome sequence of *E. endophytica* was obtained and assembled by high-throughput sequencing by Illumina HiSeq. The functional characterizations of putative genomic proteins were annotated from multiple databases, and potential effector candidates were screened using bioinformatics software. Through temporal transcriptome sequencing during early infection, the differentially expressed genes (DEGs) and effector isoenzymes in different infection stages revealed the specific pathogenetic mechanism of *E. endophytica* during *C. fissa* infection.

## 2. Materials and Methods

### 2.1. DNA Extraction and Whole-Genome Sequence of E. endophytica

The *E. endophytica* strain LS29 was isolated from a diseased seedling of the common tree *C. fissa* at Heishiding forest [5] and stored at −80 °C in an ultra-low temperature freezer (Meiling, Hefei, China). The *E. endophytica* strain LS29 was resuscitated and cultured in potato dextrose broth (PDB, Huankai Microbial, Guangzhou, China) liquid medium at 220 rpm and 28 °C for a week. Then, fungal mycelia were collected with sediment after centrifuging at 4000 rpm for 6 min and washing three times with sterile distilled water. For DNA extraction, fungal mycelia were collected, and DNA was extracted using an Ezup Column Fungi Genomic DNA Purification Kit (Sangon Biotech, Shanghai, China). After extraction, the genomic DNA was sequenced using the Illumina HiSeq sequencing platform of Novogene (Beijing, China). To improve the assembly quality, raw data were filtered and low-quality sequence fragments were removed, including (1) low-quality reads with a base mass value less than or equal to 38 and a full length over 40 bp; (2) reads with N-bases of more than 10 bp in the read sequence; and (3) reads and adapters with an overlap exceeding 15 bp threshold. Clean data were obtained to ensure the accuracy of the subsequent information analysis results. Subsequently, the clean data were spliced and assembled using SOAP denovo2 software v2.40 [29]. Different K-mer lengths (35, 47, 59, 71, 83, 95, 107, 119) were selected for assembly, in which the basic assembly data were obtained by the optimal K-mer screening, and fragments below 500 bp were filtered out to obtain the final assembly data.

### 2.2. Genomic Annotation of E. endophytica

The assembly data components were predicted and categorized into coding genes, non-coding RNA, and repeated sequences. Coding genes were predicted using *ab initio* with AUGUSTUS v3.3.2 [30]. Compared with various databases, the coding genes were annotated with the blast results under the condition of e-value ≤ 1 × 10^−5^, identity ≥ 40%, and coverage ≥ 40%. The annotated databases included Gene Ontology (GO, http://geneontology.org/, accessed on 21 December 2021) [31,32], Clusters of Orthologous Groups of proteins (COG, http://www.ncbi.nlm.nih.gov/COG/, accessed on 21 December 2021) [33], Carbohydra-Active enZYmes Database (CAZy, http://www.CAZY.org/, accessed on 21 December 2021) [34], Kyoto Encyclopedia of Genes and Genomes (KEGG, http://www.genome.jp/kegg/, accessed on 21 December 2021) [35], Non-Redundant Protein Database (Nr, https://www.ncbi.nlm.nih.gov/refseq/about/nonredundantproteins/, accessed on 21 December 2021), the Protein families database (Pfam, http://pfam-legacy.xfam.org/, accessed on 21 December 2021) [36], and the Transporter Classification Database (TCDB, http://www.tcdb.org/tcdb/, accessed on 21 December 2021) [37].

### 2.3. Effector Candidate Prediction and Pathogenic Analysis of E. endophytica

Based on the genome data, candidate-secreted effector proteins were predicted using multiple protein-related bioinformatics software. First, SignalP 5.0 (https://services.healthtech.dtu.dk/services/SignalP-5.0/, accessed on 30 October 2022) was used for screening protein sequences with N’-terminal signal peptides [38], and protein sequences that contained the membrane structure domain were removed using TMHMM 2.0 (https://services.healthtech.dtu.dk/services/TMHMM-2.0/, accessed on 30 October 2022) [39]. Simultaneously, Phobius (https://www.ebi.ac.uk/Tools/pfa/phobius/, accessed on 30 October 2022) [40] incorporated both transmembrane topology and signal peptides. The classically secreted proteins predicted by the combination of the above methods were identified as the putative secretome of *E. endophytica* strain LS29. Second, the subcellular localizations of the classically secreted proteins were predicted using Wolfpsort (https://wolfpsort.hgc.jp, accessed on 30 October 2022) and ProtComp 9.0 (http://www.softberry.com/berry.phtml?topic=protcompan&group=programs&subgroup=proloc, accessed on 30 October 2022). Third, PredGPI (https://busca.biocomp.unibo.it/predgpi/, accessed on 30 October 2022) [41] and NetGPI 1.1 (https://services.healthtech.dtu.dk/services/NetGPI-1.1/, accessed on 30 October 2022) [42] were used to predict protein sequences with a GPI modification site, and nine potential GPI modification sites were reserved. Finally, effector candidates were identified using EffectorP-fungi 3.0 (http://effectorp.csiro.au/, accessed on 30 October 2022) [43]. The databases of proteins associated with the disease included the Database of Fungal Virulence Factors (DFVF, http://sysbio.unl.edu/DFVF/, accessed on 30 October 2022) [44] and the Pathogen-Host Interactions database (PHI, http://www.phi-base.org/, accessed on 30 October 2022) [45].

### 2.4. Early Infection Experiment

To explore the molecular mechanism of the pathogenic process, a time-dependent infection experiment was carried out using *E. endophytica* strain LS29 mycelia on *C. fissa* seedlings. The seeds of *C. fissa* were surface disinfected (75% ethanol (Aladdin, Shanghai, China) 1 min, 2.63% NaClO (Aladdin, Shanghai, China) 3 min, 75% ethanol 1 min, sterile distilled water 1 min) and sowed with sterilized sand until 2 months of age before experimentation. The 2-month-old seedlings were pretreated with Hoagland’s nutrient solution (Hopebio, Qingdao, China) for 2 weeks and then transferred to sterile distilled water. The fungal mycelia were prepared according to the method described in Section 2.1. Briefly, 6 g (~200 mg dry biomass) of fungal mycelia was inoculated around the seedling roots, while the plant controls were treated with an equal weight of sterile distilled water. Fungal mycelia of equal weight were used as fungal controls (control check, CK). Mixed samples of roots and mycelia were collected at 2 h post-infection (hpi), 6 hpi, 12 hpi, 24 hpi, and 48 hpi. The mixed samples were cut into pieces with sterilized scissors and placed into 2-mL sterilized centrifuge tubes. The samples were rapidly frozen in liquid nitrogen for 5 min and kept at −80 °C for transcriptome sequencing.

### 2.5. High-Throughput Transcriptomic Sequencing

The RNA extraction and transcriptome RNA library sequencing were performed using the Illumina HiSeq sequencing platforms of Sangon Biotech (Shanghai, China). The original image data obtained were converted into raw data through CASAVA v1.8.2 Base Calling. The quality of the raw data was evaluated by FastQC v0.11.9 [46]. Trimmomatic was used to filter the raw data to ensure quality [47], which included the following: (1) removing the N-base sequences; (2) removing the junction sequences in the reads; (3) removing low-quality bases (Q value < 20); (4) removing the base of reads tails with a mass value below 20 using a sliding window method (window size of 5 bp); and (5) removing reads with a length less than 35 nt as well as their paired reads. HISAT2 v2.2.1was used to compare the quality control sequences with the genomic DNA sequences that were used as reference sequences [48], and RseQC v4.0.0was used to calculate the comparison results [49].

### 2.6. Statistical Analysis

Pearson’s correlation coefficients and principal component analysis (PCA) were used to test correlations between samples. The DEGs were analyzed by DESeq2 analysis, and genes with |log_2_ (Fold Change)| > 1 and *q*-value < 0.05 were considered DEGs.

All analyses were carried out in the R statistical environment [50] using the following packages: “ClusterProfiler” [51], “Deseq2” [52], “ggraph” [53], “ggplot2” [54], “heatmap” [55], “oaqc” [56], “psych” [57], and “vegan” [58].

## 3. Results

### 3.1. Genome Features and Completeness of E. endophytica

The draft genome of *E. endophytica* was 43 Mb in size with a G + C content of 56.65%, in which 37,673,308 bases obtained were spliced into 348 contigs and then assembled into 291 scaffolds. There was a total of 131 tRNA and 36 rRNA, 44 snRNA, and 8728 coding sequences (CDSs). The N50 length represents the sum of the lengths of all the predicted genome sequences longer than the median, which is generally used to assess the quality of the genome assembly. The Contig N50 of *E. endophytica* (>300 kb) indicated that most of the genes had relatively complete sequences, which provided a foundation for the subsequent gene prediction and functional analysis. The genome statistics are presented in Table 1.

### 3.2. Functional Annotation of the E. endophytica Genome

The whole-genome sequence of *E. endophytica* was annotated using the GO, KOG, CAZy, KEGG, NR, Pfam, and TCDB databases, the results of which have been summarized in Appendix A. By comparing the amino acid sequences of *E. endophytica* with the GO database, a total of 5747 coding genes were categorized into biological process, cellular component, and molecular function categories (Appendix A). In the biological processes category, most genes participate in metabolic processes, cellular processes, and the establishment of localizations. In the cellular component category, most genes were associated with cell parts, organelles, and macromolecular complexes. In the molecular functions category, most genes were involved in binding, catalytic activities, and transporter activities. A total of 2123 KOG annotations were obtained, for which the top seven classifications were [O] posttranslational modification, protein turnover, chaperones (217), [J] translation, ribosomal structure and biogenesis (214), [R] general function prediction only (206), [C] energy production and conversion (184), [E] amino acid transport and metabolism (154), [U] intracellular trafficking, secretion, and vesicular transport (122), and [G] carbohydrate transport and metabolism (114) (Appendix A). The genome of *E. endophytica* encoded 465 proteins homologous to CAZymes, which provided the carbohydrate enzyme family data, including 238 glycoside hydrolases (GHs), 82 glycosyl transferases (GTs), 67 auxiliary activities (AAs), 30 carbohydrate-binding molecules (CBMs), 28 carbohydrate esterases (CEs), and 20 polysaccharide lyases (PLs). To further understand the functions of the protein of *E. endophytica*, 7836 genes were annotated and assigned to five known classifications of KEGG, including metabolism, processing, processes, diseases, and systems. Translation was the most enriched gene distribution, followed by carbohydrate metabolism, amino acid metabolism, folding, sorting and degradation, and transport and catabolism in KEGG (Appendix A). With the Nr database annotation, the most matched species for the annotated genes of *E. endophytica* was *Neofusicoccum parvum* (4028), followed by *Macrophomina phaseolina* (1858) and *Diplodia seriata* (920). A total of 5747 annotation results were retrieved from the Pfam database. The FAD/NAD(P)-binding Rossmann fold superfamily (1735), P-loop containing nucleoside triphosphate hydrolase superfamily (1534), and major facilitator superfamily (554) were the main protein families in *E. endophytica*. Following TCDB annotation, 193 coding genes belonged to electrochemical potential-driven transporters, 146 coding genes were classified as primary active transporters, and 71 coding genes were channels or pores.

### 3.3. Identification of Effector Candidates and Pathogenic Annotation Based on the E. endophytica Genome

Effector candidates were predicted by SignalP, TMHMM, Phobius, Wolfpsort, ProtComp, PredGPI, NetGPI, and EffectorP (Appendix A). Of the 8728 putative proteins of *E. endophytica*, 762 were believed to carry N-terminal signal peptides using SignalP 5.0, and then, 689 were predicted to be proteins with 0/1 transmembrane helices by TMHMM 2.0. Phobius obtained 1007 predictions that met these two requirements from the proteome. A total of 681 results were retained by the combination of the three algorithms, which were identified as the putative secretome of *E. endophytica*. Subsequently, 465 expected proteins were predicted to be localized outside the organism of the fungus, combining the results from Wolfpsort (635) and ProtComp 9.0 (483) from the secretome. PredGPI and NetGPI 1.1 were used to predict the presence of GPI anchoring. The GPI-anchored secreted proteins were excluded, and 417 expected proteins without GPI-anchoring were obtained using PredGPI (425) and NetGPI 1.1 (419). Finally, 155 effector candidates were identified using EffectorP-fungi 3.0. Among the effector candidates, 128 were believed to be apoplastic effectors, 15 were regarded as cytoplasmic effectors, and the other 12 were considered to have both possibilities. Among the effector candidates of *E. endophytica*, most (71.61%) ranged in length from 100 to 300 amino acids, and one was a large effector of 556 amino acids in length (Appendix A).

According to the effector annotations, PHI-base provided 33 blasted results that contained 22 results by DFVF, which indicated that 78.71% of the effector candidates were not known to be associated with the disease. These annotated effector homologs secreted by 16 pathogens cause diseases such as root rot, leaf spot, and *Fusarium* ear blight. Based on the NCBI blast results, 29.03% of the effector candidates were identified as hypothetical proteins, and 3.23% of the effector candidates constituted orphan genes with no matches.

### 3.4. Overview of the Temporal Transcriptomic Analysis during Early Infection

Following data evaluation and quality control of the transcriptomes at 2 hpi, 6 hpi, 12 hpi, 24 hpi, and 48 hpi and CK, clean reads ranging from 32,995,466 to 50,124,544 million/sample were obtained, and the GC content of the filtered sequence was 47.57–59.48%. The proportions of filtered sequence error rates < 1% (Q20) were 98.05–98.60%, and the proportions of filtered sequence error rates < 0.1% (Q30) were 93.64–95.05% (Appendix A). The correlation heatmap indicated that the samples exhibited high inner-group similarity and inter-group separation based on Pearson’s correlation coefficients (Appendix A). The PCA showed that the 48 hpi group and other groups were separately aggregated by PC 1 (70%), and PC 2 (10%) separated the 2 hpi, 6 hpi, 12 hpi, 24 hpi, and CK groups, but not the 48 hpi group, because of the high dispersion degree of the repeated samples within the 48 hpi group (Appendix A).

### 3.5. Biosynthesizing Antibiotics to Create an Infective Environment before 24 hpi 

With the extension of infection time, the numbers of DEGs increased gradually, and the transcriptome results intensified after 24 hpi. At the 2–6, 2–12, 2–24, and 2–48 hpi stages, there were 702, 995, 1684, and 2686 upregulated genes and 546, 893, 1715, and 2994 downregulated genes (Figure 1 and Appendix A). The CK group was not considered for DEG comparison, because the control group differed significantly from all experimental groups (Appendix A).

The KEGG pathway enrichment analysis of the DEGs showed that biosynthesis of secondary metabolites was the most overrepresented pathway, followed by microbial metabolism in diverse environments and biosynthesis of antibiotics (Figure 2). Carbon metabolism, glycolysis/gluconeogenesis, starch and sucrose metabolism, and fructose and mannose metabolism were also enriched. Biosynthesis of antibiotics was upregulated before 24 hpi, and pentose and glucuronate interconversions were gradually upregulated after 12 hpi. On the contrary, ribosome biogenesis in eukaryotes, protein processing in endoplasmic reticulum, protein processing in endoplasmic reticulum, and ribosome were the most downregulated enriched pathways in the four periods (Appendix A).

### 3.6. Identification of Effective and Specific Effectors during Early Infection

The transcriptome showed that 125 effector candidates were expressed during infection. The expression of 20 effector candidates was significantly upregulated over time, and the expression of 46 effector candidates was significantly downregulated (Figure 3A). Some effector candidates were significantly upregulated at 2–24 hpi but significantly downregulated at 24–48 hpi, while some genes showed the opposite (Figure 3B,C). Importantly, the top 10 effector candidates with high gene expression included A0194 (hypothetical protein), A0439 (Eliciting plant response protein 1, Epl1), A1948, A2703, A4087, A4417, A5728, A6540 (hypothetical protein), A8041 (Necrosis-inducing secreted protein 1, Nis1), and A8732 (Ubiquitin 3 binding protein, But2) (Figure 3D). The gene annotations of the top 10 expressed effectors were summarized in Appendix A. The A0439 proteins were highly expressed without significant change, and the A8041 proteins were significantly upregulated at an extremely high expression level. Of these hypothetical proteins with high gene expression, the expression of A0194 and A5728 was significantly upregulated. The A4087 and A4417 proteins were expressed and then significantly downregulated at 48 hpi. In addition, A1948, A2703, and A6540 were expressed without any significant difference over time.

Among the known or putative effectors that matched with PHI-base and DFVF, A8077, A3579 (pectate lyase, PL3_2), and A3834 (endopolygalacturonase, GH28) were significantly and persistently upregulated during the experiment. In addition, A8509 (cellobiohydrolase, GH7), A0949 (chitin deacetylase, CBM18), A0448 (chitinase, GH18), A5327 (lipase class 3), A0325 (metalloprotease), A5511 (pectate lyase, PL3_2), and A2151 (pectinesterase, CE8) were upregulated until 12–24 hpi and then significantly downregulated.

### 3.7. Comparative Analysis of Isoenzymes Reveals Specific Pathogen-Related Genes during Early Infection

The isoenzymes were obtained from the secretome of *E. endophytica* based on effector candidates and were analyzed by a general linear regression model. Isoenzyme analysis showed that A4173 (cellobiohydrolase), A5330 (cell wall protein), A6987 (endoglucanase), A1416, A5475, A5601, A5692 (lipase), A4055, A8077, A3579 (pectate lyase), A3705, and A3706 (rhamnogalacturonase) were significantly upregulated at high expression levels (Figure 4).

Although there were no significant differences across infection stages, A8720 (carboxypeptidase), A5369, A3278, A3788 (lipase), and A7584 (chitin deacetylase) still had high levels of expression. Other isoenzymes expressed at low levels were downregulated or were not significantly differentially regulated.

## 4. Discussion

The whole-genome annotation suggested that most of the genes of *E. endophytica* were involved in the translation, processing, and transportation of proteins, and many genes were associated with carbohydrate and amino acid metabolism, which are vital for pathogen infection and survival in the host. More than half of the genes had not been annotated. There are few studies on *Endomelanconiopsis* and its related genera. The best genome-matched species in the Nr database is *N. parvum*, though less than half of the CDSs of *E. endophytica* matched with those of *N. parvum. N. parvum* is a pathogen causing black leaf spot disease on *Geodorum eulophioides* in China [59], while *E. endophytica* causes root rot disease in *C. fissa*. These two pathogens belong to the family Botryosphaeriaceae and may have similar pathogenic mechanisms that are worthy of further comparative research. We obtained 681 genes of the secretome and 155 genes of effector proteins through bioinformatics prediction. Effectors play an important role in the pathogen infection process of host plants, including recognition, adsorption, infection, colonization, and reproduction between pathogens and host plants. The effector prediction indicated that most of the effectors of *E. endophytica* worked on the extracellular matrix of plants during infection. The pathogenicity-related annotation of effector candidates indicated that the known effector genes of *E. endophytica* were mainly enzymes, such as pectate lyases, chitinases, pectinesterases, and endopolygalacturonases, which are involved in hydrolyzing the pectin and cellulose of the plant cell wall. This corresponds to the CAZyme annotation of *E. endophytica,* which was found to contain a large number of genes homologous to GH16, GH3, and AA3, which interact collaboratively on plant cell walls and cellulose as the first defense line of the plant immune system [12]. Unfortunately, more than 79% of effector proteins had unknown functions, and some of the effector proteins did not even have homologs. Though the blasted genes revealed the possible pathogenic mechanism of *E. endophytica*, the remaining genes may shed light on the host-specificity and potential unknown pathogenic mechanism of *E. endophytica*.

The temporal transcriptome herein provided us with further evidence about pathogenesis. Unfortunately, the CK group was not included in the DEG comparison because of its significant difference. The fungal CK group samples were collected on a PDA solid medium, while the other group samples were mixtures of root and fungi were collected in a liquid medium. The fungal content and culture conditions influenced the gene expression, but the CK group could be used as a reference for assessing expression. Also, fungal transcriptome expression in the plant control group was too small to be considered, so it was not included in the data analysis. With the extension of infection time, some DEGs showed a significant opposing trend before and after 24 hpi, suggesting that some genes were temporarily out of use, while others were re-invoked. This also indicated that 24 hpi was a key time point at which *E. endophytica* had moved on to the next stage of pathogenesis.

The KEGG pathway enrichment analysis of upregulated genes revealed that biosynthesis of secondary metabolites was one of the methods for fungal invasion. Secondary metabolites are not essential for growth, but some of them are considered effectors in disease establishment and the induction of necrotrophy, such as polyketides, non-ribosomal peptides, hybrid polyketide–non-ribosomal peptides, and terpenes [60]. T-toxin, victorin, and HC-toxin are used by the host-specific necrotrophic fungi of *Cochliobolus* spp. to establish disease in maize [61]. Biosynthesis of antibiotics was no longer significantly upregulated after 24 hpi, suggesting that the fungi secreted antibiotics for the purpose of inhibiting other microorganisms from occupying the plant territory when the fungi met the host plant. Antibiotics are produced to destroy the competing organisms present in the same habitat, and cephalosporin is a well-known antibiotic that is produced by *Acremonium* spp. against both Gram-positive and negative bacteria [62]. Other KEGG-enriched upregulated pathways indicated that the fungi activated multiple pathways involved in energy metabolism and effective colonization to facilitate the infection process. The enriched KEGG pathways of the upregulated and downregulated genes also demonstrated the energy allocation of the fungus during infection. Understanding the role of metabolic pathways may require additional metabolomic studies to explore specific metabolites during infection.

Our DEG expression analysis of effector candidates indicated that more than 80% of effectors participated in infection. However, many effectors were secreted to be involved in host recognition before 24 hpi, but only a few effectors successfully recognized and effectively worked during invasion after 24 hpi. The expression of A0439 (Epl1) was always high without significant upregulation or downregulation. A cerato-platanin protein called Epl1 can be secreted by *Colletotrichum falcatum* as a virulence factor or elicitor to induce hypersensitive response (HR) -like cell death in *Nicotiana tabacum* [63], indicating that *E. endophytica* had been in the active state of attack of the host for a long time. The A8041 (Nis1) proteins were persistently expressed and significantly upregulated to an extremely high level. The Nis1 protein is thought to be a core effector that targets BAK1 (brassinosteroid-insensitive 1 (BRI1) associated receptor kinase 1) and BIK1 (botrytis-induced kinase 1), interfering with functions for immune activation upon pathogen recognition, and is conserved in filamentous fungi in Ascomycota and Basidiomycota [64]. Presumably, A8041 (Nis1) is believed to act as a key effector of *E. endophytica* to suppress the PAMP-triggered immunity of *C. fissa*. Pectate lyase is a hydrolytic enzyme that cleaves α-1,4-polygalacturonic acid and releases pectin products [65], and endopolygalacturonase is an enzyme required for the hydrolysis of pectin [66]. Therefore, A8077, A3579, and A3834 are expected to play a significant role in plant cell wall degradation, which is key in the first line of defense of a plant. There were many proteins without annotations that were also found to play an important role in the pathogenic mechanism of *E. endophytica*. Of the hypothetical proteins, it is possible that A1948, A2703, and A6540 were general effectors for conventional infection, and A0194 and A5728 were effective during infection, whereas A4087 and A4417 were less useful or only worked in the early 24 h infection stage when they attacked the seedlings of *C. fissa*. There were 20 significantly upregulated effector candidates containing many hypothetical proteins that may be important for the host-specific infection of *E. endophytica* on *C. fissa*.

Predictions of effector candidates based on machine learning are not accurate, so the secreted isoenzymes of effector candidates were considered. Isoenzymes catalyze the same chemical reaction but differ in amino acid composition, intracellular location, and physiological role [67]. In our isoenzyme analysis, several effector proteins with high expression and their isoenzymes were compared and analyzed. These enzymes included carboxypeptidases, cellobiohydrolases, cell wall proteins, chitin deacetylases, endoglucanases, lipases, pectate lyases, ribonucleases, and rhamnogalacturonases, in which pectate lyases were previously introduced. Most of them are CWDEs, including cellobiohydrolases, chitin deacetylases, endoglucanases, pectate lyases, rhamnogalacturonases, and other cell wall proteins. Carboxypeptidases are involved in peptide biosynthesis and in hydrolyzing extracellular proteins or polypeptides, which play a vital role in stress response, growth, development, and pathogen defense [68,69]. Effector lipase FGL1 is secreted by *Fusarium graminearum* to release linoleic and α-linolenic acids from wheat, which suppress innate immunity-related callose biosynthesis [70]. VdRTX1, a ribonuclease of *Verticillium dahlia*, was found to translocate into the plant nucleus to modulate immunity. Its homologs are widely distributed in fungi [71]. For CWDEs, cellobiohydrolase is an enzyme that disrupts the chain end of cellulose and releases glucose [72]. Chitin deacetylase is secreted during infection and the early growth phase in the host, which alters the physicochemical properties of the cell wall to evade recognition by plant chitinases, thereby reducing the elicitation of plant defenses [73,74]. MoChi1, an extracellular chitinase of *M. oryzae*, suppressed reactive oxygen species in rice cells [75]. Lipase is an esterase that catalyzes the hydrolysis of long-chain triglycerides into free fatty acids and glycerol and may be a virulence factor for fungi [76]. Endoglucanases are suggested to contribute to the hydrolysis of cellulose compounds by other cellobiohydrolases. The apoplastic effector PsXEG1 is an endoglucanase that destroys plant cell walls in the *Phytophthora sojae*–soybean interaction. *N*-glycosylation is employed as the shield to PsXEG1 from GmAP5 degradation [77,78]. Rhamnogalacturonan is a plant cell wall compound that can be hydrolyzed by rhamnogalacturonase, but it has received little attention [79].

We found that not all disease-related proteins had significantly upregulated expression. Many isoenzymes were upregulated before 24 hpi and downregulated after 24 hpi, which were considered general effectors in plant recognition. However, the expression of only a few proteins was upregulated to high levels to play an important role during infection, and these were considered specific effectors during *C. fissa* infection. Therefore, A4173 (cellobiohydrolase), A5330 (cell wall protein), A6987 (endoglucanase), A1416, A5475, A5601, A5692 (lipase), A4055, A8077, A3579 (pectate lyase), A3705, and A3706 (rhamnogalacturonase) were considered as specific effectors of *E. endophytica* to specifically destroy the defense system of *C. fissa*.

As specific effectors, A4173, A5330, A6987, A4055, A8077, A3579, A3705, and A3706 hydrolyze plant wall cell compounds, such as cellulose, pectin, and rhamnogalacturonan, while A1416, A5475, A5601, and A5692 contribute to the release of fatty acids, thereby interfering with plant immunity. In addition, A8041 acts as a core effector interfering with the pathogen recognition of plants, and A7584 masks fungal chitin to circumvent reorganization by plants. Taken together, *E. endophytica* has a well-divided effector system for achieving host-specific infection. Indeed, the functions of these proteins await further verification, as do those effectors whose functions are unknown.

## 5. Conclusions

In this study, we examined the specific pathogenesis of *E. endophytica* on *C. fissa* in the early infection stage. The whole-genome sequence of *E. endophytica* was obtained and annotated, which provided insight into the genetic features linked to pathogenicity and virulence. A total of 155 effector candidates were predicted based on the characteristics of the effectors, and most of them were considered to be involved in attacking the cell walls of host plants. The transcriptomes revealed many details about the mechanisms of fungal pathogenesis, with 3284 genes that were upregulated and 3626 genes that were downregulated. We found that 24 hpi was a key time point in the process of *E. endophytica* pathogenesis, and some genes showed opposing expression regulation trends before and after 24 hpi. As a specific pathogen, *E. endophytica* was more focused on repressing other microbes by producing antibiotic substances before 24 hpi. The differential expression of isoenzymes revealed that only a few effectors played an important role during the early infection of *C. fissa*, which were considered specific effectors. These specific effectors were found to be involved in hydrolyzing the compounds of plant cell walls and releasing fatty acids after 24 hpi, while other important effectors worked at altering fungal chitin to evade recognition by the plant. In conclusion, our results provide the first genome and temporal transcriptome of *E. endophytica* for genetic structures, effector compounds, and host-specific pathogenesis.

## Figures and Tables

**Figure 1 jof-09-01040-f001:**
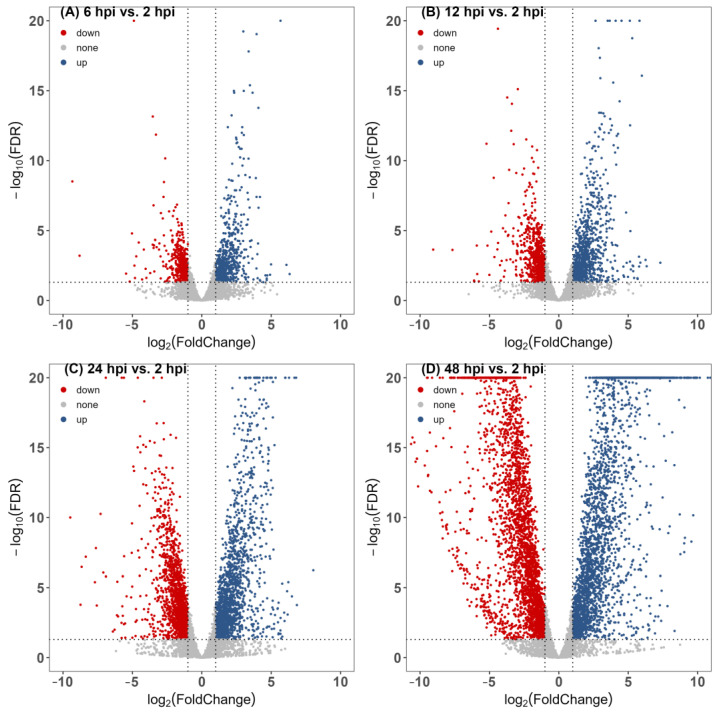
Differential gene volcano map of gene expression during the infection process of *Castanopsis fissa* by *Endomelanconiopsis endophytica*. (**A**) 6 hpi vs. 2 hpi, (**B**) 12 hpi vs. 2 hpi, (**C**) 24 hpi vs. 2 hpi, (**D**) 48 hpi vs. 2 hpi. The results are based on DESeq2 analysis. Genes with |log_2_ (Fold Change)| > 1 and *q*-value < 0.05 were considered differentially expressed. Red points represent downregulated genes, blue points represent upregulated genes, and gray points represent no significant difference in gene expression.

**Figure 2 jof-09-01040-f002:**
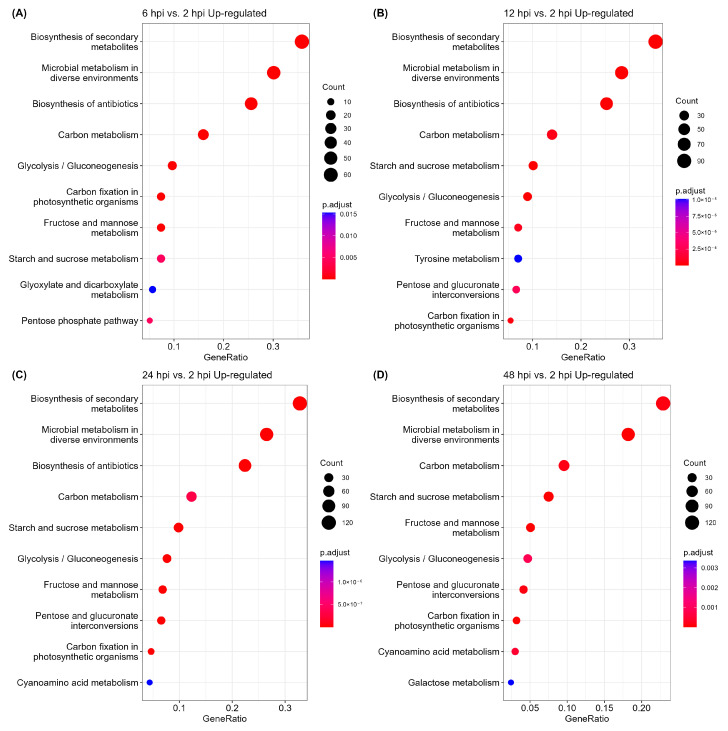
Pathway enrichment analysis of the upregulated genes during the infection process of *Castanopsis fissa* by *Endomelanconiopsis endophytica*. (**A**) 6 hpi vs. 2 hpi, (**B**) 12 hpi vs. 2 hpi, (**C**) 24 hpi vs. 2 hpi, (**D**) 48 hpi vs. 2 hpi. The top 10 Kyoto Encyclopedia of Genes and Genomes (KEGG) upregulated pathways are shown, with the abscissa representing the gene ratio, while the ordinate represents the names of the KEGG pathways. The depth of the color represents the adjusted *p*-value. The size of the circle on the graph represents the number of genes.

**Figure 3 jof-09-01040-f003:**
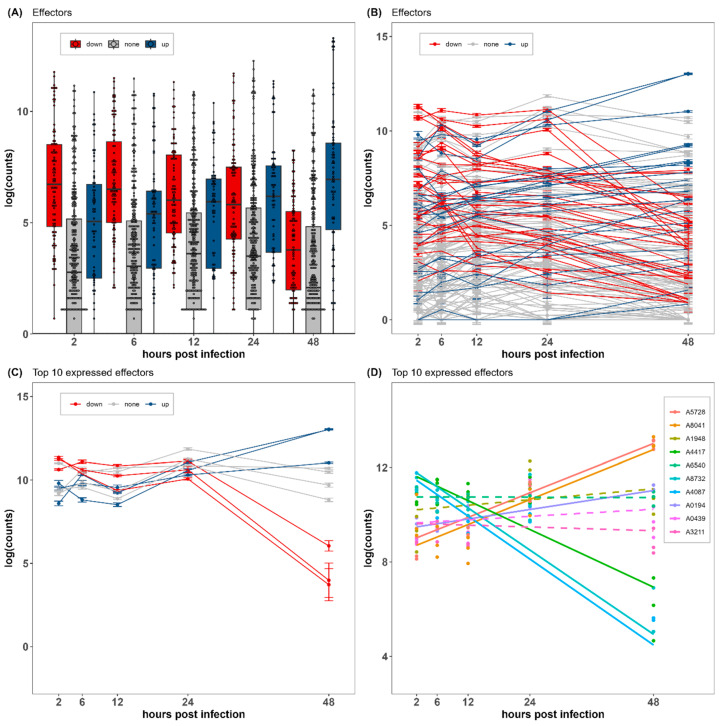
Gene expression of effectors in *Endomelanconiopsis endophytica*. (**A**) Boxplot of all effectors, line charts of (**B**) effectors and (**C**) top 10 expressed effectors, and (**D**) general linear regression of top 10 expressed effectors. The abscissa represents hours post-infection, and the ordinate represents log (counts of genes). (**A**–**C**) Red represents downregulated genes, blue represents upregulated genes, and gray represents genes with no significant differential expression. (**D**) The color represents the name of the genes, the solid line represents *p* < 0.05, and the dashed line represents *p* ≥ 0.05.

**Figure 4 jof-09-01040-f004:**
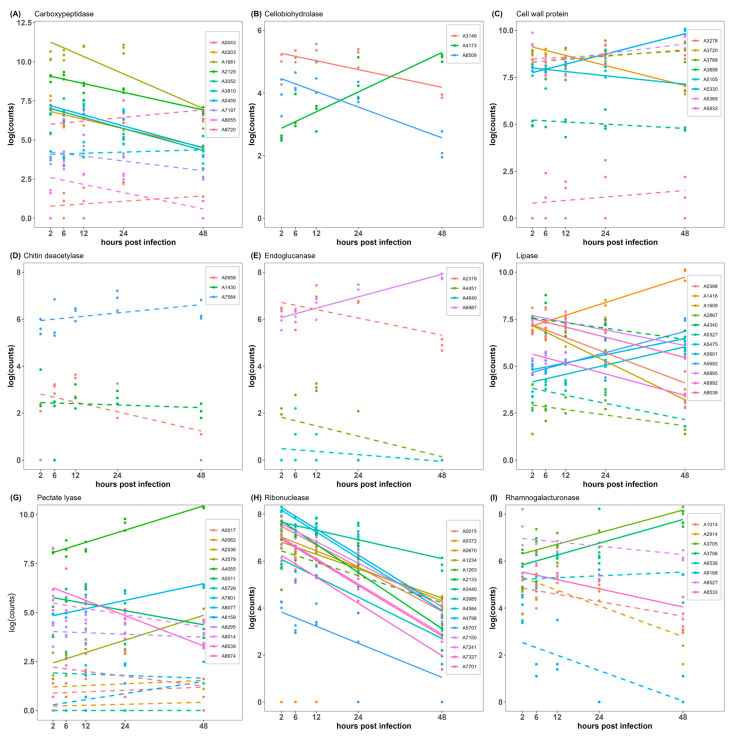
Gene expression and general linear regression of pathogenic isoenzymes in *Endomelanconiopsis endophytica*. (**A**) Carboxypeptidase, (**B**) cellobiohydrolase, (**C**) cell wall protein, (**D**) chitinase, (**E**) endoglucanase, (**F**) lipase, (**G**) pectate lyase, (**H**) ribonuclease, and (**I**) rhamnogalacturonase. The color represents the name of the genes, the solid line represents *p* < 0.05, and the dashed line represents *p* ≥ 0.05.

**Table 1 jof-09-01040-t001:** Genome features of the assembled *Endomelanconiopsis endophytica*.

Assembly Name	*Endomelanconiopsis endophytica* LS29
Assembly size (Mb)	43
K-mer Depth	26.77
Sequence GC (%)	56.65
Total Num (>500 bp) of scaffolds	291
N50 Length (bp)	551,176
N90 Length (bp)	126,219
Max Length (bp)	1,466,726
Min Length (bp)	503
Total Num (>500 bp) of contigs	348
N50 Length (bp)	332,028
N90 Length (bp)	88,513
Max Length (bp)	1,466,726
Min Length (bp)	468

## Data Availability

Relevant data are available in the online Appendix A and Dryad Digital Repository https://doi.org/10.5061/dryad.ghx3ffbvq.

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
