# Peer review of "Host Recognition and Specific Infection of Endomelanconiopsis endophytica during Early Infection"

_jof, 2023, doi:10.3390/jof9101040_

Round 1
Reviewer 1 Report
The manuscript by Yan Xie and co-authors entitled “Host recognition and specific infection of Endomelanconiopsis endophytica during early infection” (submitted to the “Journal of Fungi”) provides the novel and interesting information on revealing of the molecular mechanisms involved in host-specific pathogenesis during the interactions between E. endophytica and C. fissa in the early infection stage.
In this study, the researchers presented a 43-Mb whole-genome sequence of Endomelanconiopsis endophytica strain LS29, a host-specific pathogen of the common subtropical tree Castanopsis fissa. Its genome annotations were described, and its effector candidates were identified. By performing temporal transcriptome sequencing of E. endophytica on C. fissa during early infection, the authors found that E. endophytica repressed other microbes in order to attack the tissue of the host by producing antibiotics earlier than 24 hours post-inoculation (hpi). Simultaneously, a variety of effectors were secreted to recognize the host plant, but most of them showed a significantly opposing expression regulation trend after 24 hpi, indicating that 24 hpi represents a key time point between host recognition and specific infection. A comparison of isoenzymes showed that only a few effectors were identified as specific effectors, which were involved in hydrolyzing the compounds of the plant cell wall and releasing fatty acids during the early infection of C. fissa. This work provided the first genome and temporal transcriptome of E. endophytica for genetic structures, effector compounds, and host-specific pathogenesis.
In my opinion, this manuscript should be accepted for publication in “Journal of Fungi”.
The manuscript is written in a good scientific language; the quality of English language is good
Author Response
Thank you very much for your positive comments.
Reviewer 2 Report
Xie et al. nicely presented their work in this article. This article is informative and enjoyable. I highly recommend the manuscript be accepted after minor revision.
Here are my comments
Line 47- ": one" to ": One"
Line 112- "refrigerator" to "freezer"
Line 115- "distilled water" to "sterile distilled water"
Line 459- ".For CWDEs" to ".(space)For(single space)CWDEs"
Line 492- "In this study, we examined the interactions between E. endophytica and C. fissa"- the statement is incorrect as this study examined E. endophytica only. Of course, the authors can add plant-side data, but I think they want to publish another article with the plant side.
The limitation of transcriptome analysis using "2hr" as a standard is understandable as obtaining RNA-Seq data of 0hr post-inoculation in the fungal pathogen side is problematic. For future papers on the plant side, however, I highly recommend conducting a 2-hour mock vs. 2-hour inoculated/6-hour mock vs. 6-hour inoculated... so on because the circadian cycle on the plant side can change transcriptome levels.
Another suggestion can be making a table (or a few tables) listing the top 10 candidates in each analysis.
I can't say it is innovative or novel, but it is a good paper.
Just minor errors.
Author Response
Xie et al. nicely presented their work in this article. This article is informative and enjoyable. I highly recommend the manuscript be accepted after minor revision.
Response: Thank you very much for your positive comments.
Here are my comments
Q1. Line 47- ": one" to ": One"
Response: Revised.
Q2. Line 112- "refrigerator" to "freezer"
Response: Revised.
Q3. Line 115- "distilled water" to "sterile distilled water"
Response: Revised.
Q4. Line 459- ".For CWDEs" to ".(space)For(single space)CWDEs"
Response: Revised.
Q5. Line 492- "In this study, we examined the interactions between E. endophytica and C. fissa"- the statement is incorrect as this study examined E. endophytica only. Of course, the authors can add plant-side data, but I think they want to publish another article with the plant side.
Response: Thank you very much for your comments and we revised the sentence to “In this study, we examined the specific pathogenesis of E. endophytica on C. fissa in the early infection stage.”
Q6. The limitation of transcriptome analysis using "2hr" as a standard is understandable as obtaining RNA-Seq data of 0hr post-inoculation in the fungal pathogen side is problematic. For future papers on the plant side, however, I highly recommend conducting a 2-hour mock vs. 2-hour inoculated/6-hour mock vs. 6-hour inoculated... so on because the circadian cycle on the plant side can change transcriptome levels.
Response: Thank you very much for your suggestions.
Q7. Another suggestion can be making a table (or a few tables) listing the top 10 candidates in each analysis.
Response: Thank you very much for your suggestions and we have added a table (Table S3) in the supplementary.
Reviewer 3 Report
The manuscript "Host recognition and specific infection of Endomelanconiopsis endophytica during early infection" (jof-2662021) is a study that examines the early stages of infection of a fungus and its specific plant host, using genomics, comparative transcriptomics, and isoenzyme comparison.
The technical work is very well executed, but the introduction is weak. The authors continue previous work (Cheng 2020) and state “However, the pathogenic mechanism between host-specific pathogens and host plants remains unclear” (lines 34-35). But this refers to the mechanism of this specific fungus (as reaffirmed in lines 106-107: “the specific pathogenetic mechanism of E. endophytica during C. fissa infection”), and as it is worded it seems like a general statement. It must be corrected because there are articles that describe similar mechanisms in other fungi. Some of them are even cited later in the manuscript. Other related articles not mentioned are:
Jiming Li, Ben Cornelissen, Martijn Rep,
Host-specificity factors in plant pathogenic fungi,
Fungal Genetics and Biology, Volume 144, 2020, 103447, https://doi.org/10.1016/j.fgb.2020.103447
Zhang S, Li C, Si J, Han Z, Chen D.
Action mechanisms of effectors in plant-pathogen interaction
Int J Mol Sci. 2022 Jun 17;23(12):6758. https://doi.org/10.3390/ijms23126758
Takashi Tsuge, Yoshiaki Harimoto, Kazuya Akimitsu, Kouhei Ohtani, Motoichiro Kodama, Yasunori Akagi, Mayumi Egusa, Mikihiro Yamamoto, Hiroshi Otani
Host-selective toxins produced by the plant pathogenic fungus Alternaria alternata
FEMS Microbiology Reviews, Volume 37, Issue 1, January 2013, Pages 44–66, https://doi.org/10.1111/j.1574-6976.2012.00350.x
Other aspects that may help improve the manuscript are listed below:
- Line 25. Keywords are usually ordered alphabetically. “effector” is disordered.
- Lines 134-162. Perhaps it is an editorial question, will the hyperlinks be included in the final article? Line 166. "sterilized". I think “surface disinfected” is more correct. The same word in the next line, on the contrary, seems correct to me.
- Line 195. Reference “R Development Core Team, 2021” is not in the References section.
- Line 240: rossmann. Please, capitalize: Rossmann.
- All the figures. Please, adjust the figures to the center.
- Line 459. “(Yin et al. 2022).For”. The text lacks an extra space after the dot.
References section:
- There are some references in this section that do not appear in the text: lines 529-531, 568-569, 586-588, 649-650, and 680-681.
- There are some possibly incomplete references. Please, review all the section, but especially lines 547-548, 549, lines 550-551, 604, 605-606, 632, 638,
- No reference has the biological names written in italics!!!
- There is a lack of homogeneity in the style of some of the references. For example, in the reference on lines 572-574, the name of the journal is abbreviated. In the following reference, and in almost the entire text, it is complete. They must unify the style according to the journal’s instructions for authors.
The English language should be carefully reviewed. Examples:
- Line 75. “At preseent”.
- Lines 88-89. “Through combined with transcriptome analysis, more effective experimental evidence of effector candidates has been provided”.
- Lines 218-220. “In the biological processes category, metabolic processes, cellular processes, and establishment of localizations were the dominant terms”.
Author Response
Q1. The manuscript "Host recognition and specific infection of Endomelanconiopsis endophytica during early infection" (jof-2662021) is a study that examines the early stages of infection of a fungus and its specific plant host, using genomics, comparative transcriptomics, and isoenzyme comparison.
The technical work is very well executed, but the introduction is weak. The authors continue previous work (Cheng 2020) and state “However, the pathogenic mechanism between host-specific pathogens and host plants remains unclear” (lines 34-35). But this refers to the mechanism of this specific fungus (as reaffirmed in lines 106-107: “the specific pathogenetic mechanism of E. endophytica during C. fissa infection”), and as it is worded it seems like a general statement. It must be corrected because there are articles that describe similar mechanisms in other fungi. Some of them are even cited later in the manuscript. Other related articles not mentioned are:
Jiming Li, Ben Cornelissen, Martijn Rep,
Host-specificity factors in plant pathogenic fungi,
Fungal Genetics and Biology, Volume 144, 2020, 103447, https://doi.org/10.1016/j.fgb.2020.103447
Zhang S, Li C, Si J, Han Z, Chen D.
Action mechanisms of effectors in plant-pathogen interaction
Int J Mol Sci. 2022 Jun 17;23(12):6758. https://doi.org/10.3390/ijms23126758
Takashi Tsuge, Yoshiaki Harimoto, Kazuya Akimitsu, Kouhei Ohtani, Motoichiro Kodama, Yasunori Akagi, Mayumi Egusa, Mikihiro Yamamoto, Hiroshi Otani
Host-selective toxins produced by the plant pathogenic fungus Alternaria alternata
FEMS Microbiology Reviews, Volume 37, Issue 1, January 2013, Pages 44–66, https://doi.org/10.1111/j.1574-6976.2012.00350.x
Response: Thank you very much for your comments. We rephrased the sentence as “the pathogenesis of host-specific pathogen E. endophytica on C. fissa remains unclear.” We have also added the references you suggested (line 33-34, line 64-66, and line 76-77).
Q2. - Line 25. Keywords are usually ordered alphabetically. “effector” is disordered.
Response: Thank you very much for your suggestions and We have changed the keywords to “comparative transcriptomics; effector; genomics; specific infection; whole-genome sequence”
Q3. - Lines 134-162. Perhaps it is an editorial question, will the hyperlinks be included in the final article?
Response: Thank you very much for your comments and this may be decided by the Editor.
Q4. Line 166. "sterilized". I think “surface disinfected” is more correct. The same word in the next line, on the contrary, seems correct to me.
Response: Thank you very much for your comments. Seeds were surface disinfected, and sands were sterilized by autoclave sterilizer. We have corrected.
Q5.- Line 195. Reference “R Development Core Team, 2021” is not in the References section.
Response: Thank you very much for your comments and we have added the citation.
Q6. - Line 240: rossmann. Please, capitalize: Rossmann.
Response: Revised.
Q7.- All the figures. Please, adjust the figures to the center.
Response: Revised.
Q8. - Line 459. “(Yin et al. 2022).For”. The text lacks an extra space after the dot.
Response: Revised.
Q9.- There are some references in this section that do not appear in the text: lines 529-531, 568-569, 586-588, 649-650, and 680-681.
Response: Thank you very much for your comments. Somethings wrong while we used Endnote. We have revised in the main text.
Q10.- There are some possibly incomplete references. Please, review all the section, but especially lines 547-548, 549, lines 550-551, 604, 605-606, 632, 638,
Response: Thank you very much for your comments and we have revised in the main text.
Q11.- No reference has the biological names written in italics!!!
Response: Thank you very much for your comments and we have revised in the main text.
Q12.- There is a lack of homogeneity in the style of some of the references. For example, in the reference on lines 572-574, the name of the journal is abbreviated. In the following reference, and in almost the entire text, it is complete. They must unify the style according to the journal’s instructions for authors.
Response: Thank you very much for your comments and we have revised.
Q13. The English language should be carefully reviewed. Examples:
- Line 75. “At preseent”.
- Lines 88-89. “Through combined with transcriptome analysis, more effective experimental evidence of effector candidates has been provided”.
- Lines 218-220. “In the biological processes category, metabolic processes, cellular processes, and establishment of localizations were the dominant terms”.
Response: We apologize for these errors and have revised.